# 46,XX Testicular Disorder of Sex Development (DSD): A Case Report and Systematic Review

**DOI:** 10.3390/medicina55070371

**Published:** 2019-07-12

**Authors:** Marco Terribile, Marco Stizzo, Celeste Manfredi, Carmelo Quattrone, Francesco Bottone, Dario Ranieri Giordano, Giuseppe Bellastella, Davide Arcaniolo, Marco De Sio

**Affiliations:** 1Urology Unit, Department of Woman Child and of General and Specialist Surgery, University of Campania “Luigi Vanvitelli”, 80131 Naples, Italy; 2Division of Endocrinology and Metabolic Diseases, Department of Advanced Medical and Surgical Sciences, University of Campania “Luigi Vanvitelli”, 80131 Naples, Italy

**Keywords:** 46,XX, XX male syndrome, disorders of sex development, De la Chapelle, infertility, hypogonadism

## Abstract

*Background and objectives:* XX male syndrome is part of the disorders of sex development (DSD). The patients generally have normal external genitalia and discover their pathology in adulthood because of infertility. There are no guidelines regarding XX male syndrome, so the aim of our study was to evaluate the literature evidence in order to guide the physicians in the management of these type of patients. *Materials and Methods:* We performed a systematic review of the available literature in September 2018, using MEDLINE, Web of Science, Embase and Google Scholar database to search for all published studies regarding XX male syndrome according to PRISMA guidelines. The following search terms were used: “46 XX male”, “DSD”, “infertility”, “hypogonadism”. *Results:* After appropriate screening we selected 37 papers. Mean (SD) age was 33.14 (11.4) years. Hair distribution was normal in 29/39 patients (74.3%), gynecomastia was absent in 22/39 cases (56.4%), normal testes volume was reported in 0/14, penis size was normal in 26/32 cases (81.2%), pubic hair had a normal development in 6/7 patients (85.7%), normal erectile function was present in 27/30 cases (90%) and libido was preserved in 20/20 patients (100%). The data revealed the common presence of hypergonadotropic hypogonadism. All patients had a 46,XX karyotype. The sex-determining region Y (*SRY*) gene was detected in 51/57 cases. The position of the *SRY* was on the Xp in the 97% of the cases. *Conclusions:* An appropriate physical examination should include the evaluation of genitalia to detect cryptorchidism, hypospadias, penis size, and gynecomastia; it is important to use a validated questionnaire to evaluate erectile dysfunction, such as the International Index of Erectile Function (IIEF). Semen analysis is mandatory and so is the karyotype test. Abdominal ultrasound is useful in order to exclude residual Müllerian structures. Genetic and endocrine consultations are necessary to assess a possible hypergonadotropic hypogonadism. Testicular sperm extraction is not recommended, and adoption or in vitro fertilization with a sperm donor are fertility options.

## 1. Introduction

Disorders of sex development (DSD) is used as an umbrella term [1] for various rare conditions that are characterized by an incongruence of chromosomal, gonadal, and genital sex development. XX male syndrome is part of DSD, so it is also called 46,XX testicular disorder of sex development. Patients generally have normal external genitalia and discover their pathology in adulthood because of infertility. Typical features of this condition are: 46,XX karyotype, normal male phenotype, small testes, azoospermia, and hypergonadotropic hypogonadism [2].

It is a rare disease occurring in about 1:20,000 males [3]. The first case was described in 1964 by De la Chapelle [4]; since then, several other cases were reported. In this study we present the case report of a 36-year-old man suffering from XX male syndrome, presented as infertility, and perform a systematic review of the available evidences about the topic.

### Case Report

A 36-year-old man came to our clinic complaining about infertility; he engaged in regular, unprotected sexual intercourse during the last 20 months without his wife becoming pregnant. His 30-year-old wife underwent a gynecological consultation and no remarkable diseases were diagnosed. He reported no familiar history of endocrine diseases, genetic syndromes or infertility and his medical history revealed only carpal tunnel release surgery; furthermore, no history of testicular trauma or cryptorchidism was present. The patient’s job did not expose him to radiation or cytotoxic agents, and he did not take any medication. He had normal libido, good erectile function (International Index of Erectile Function (IIEF)-5 score: 22 points), normal morning erections, and no genital or urinary troubles. The patient complained of mild asthenia, impaired concentration, and breast growth in the last 2 years.

The height and weight of patient were 165 cm and 74 kg, respectively, with a BMI (body mass index) of 27.1 kg/m^2^), sagittal abdominal diameter of 29 cm, and his blood pressure was 110/70 mmHg. He presented sparse body hair and bilateral gynecomastia (grade II). The genital examination showed symmetrical male genitalia, stretched penis length of 8 cm, small testes (both 6 mL), and sparse pubic hair (Tanner stage II). No clinical varicocele was found. Digital rectal examination revealed a normal prostate gland. Standard abdominal ultrasound showed no significant disorders. Normal prostate gland and normal seminal vesicles, with no Müllerian derivates, were found with pelvic ultrasound. No varicocele was diagnosed with testicular ultrasound.

Hormone analysis revealed hypergonadotropic hypogonadism: follicle-stimulating hormone (FSH) and luteinizing hormone (LH) were 24.7 mIU/mL (1–13 mIU/mL) and 9.4 mIU/mL (1–9 mIU/mL) respectively, whereas early morning total testosterone (TT) was 235 ng/dL (300–1200 ng/dL) and free testosterone (FT) calculated by formula was 3.5 pg/mL (9–30 pg/mL). Estradiol (E2) and prolactin (PRL) levels were 14 pg/mL (10–40 pg/mL) and 12.2 ng/mL (4–23 ng/mL) respectively, prostate-specific antigen (PSA) was 0.6 ng/mL, blood sugar was 88 mg/dL, total cholesterol was 213 mg/dL (<200 mg/dL), hematocrit was 44% (41–50%) and hemoglobin was 15.4 g/dL (14–17.5 g/dL). The laboratory parameters were confirmed by a second dosage. 

The semen analysis, according to the guidelines of the World Health Organization (WHO) Laboratory Manual for the Examination and Processing of Human Semen (5th edition), was performed after four days of abstinence and showed normal ejaculated volume (2.4 mL) and azoospermia after centrifugation. The semen collection was performed in the laboratory and it was analyzed within the following 30 min by two expert biologists. A second and a third sample confirmed similar values. 

Karyotyping was performed on peripheral blood lymphocytes and showed a 46,XX karyotype. Fluorescent in situ hybridization (FISH) was carried out using the Vysis *SRY* probe revealing the sex-determining region Y (*SRY*) on the short (p) arm of the X chromosome. The patient underwent genetic consultation that confirmed the diagnosis of 46,XX (*SRY*-positive) DSD. A testicular biopsy was proposed to get a histological diagnosis, but the patient refused. Artificial insemination with sperm donation and psychological support were offered to the couple, and the patient is on clinical and laboratoristic follow-up.

## 2. Materials and Methods

### 2.1. Search Strategy

We performed a systematic review of the available literature in September 2018, using MEDLINE, Web of Science, Embase, and Google Scholar database to search for all published studies regarding XX male syndrome. The following search terms were used: “46 XX male”, “DSD”, “infertility”, “hypogonadism”. 

### 2.2. Inclusion and Exclusion Criteria

We included papers that met the following criteria: English language, human studies, adult patients (≥18 years old), full-text availability, completeness of clinical and laboratory data. No filters were applied for the date of publication.

Case reports of patients of pediatric age group and studies not having primary data (i.e., reviews not including case report, commentaries, and letters) were excluded, however they were examined to include any possible relevant citations. 

### 2.3. Data Extraction

Reference lists in relevant studies were used to search for additional studies. After a first screening based on study titles and abstracts, all selected papers were assessed based on the full-text to choose the relevant publications to be included in the analysis. 

Two authors carried out this review independently (M.T., M.S.), and disagreements regarding the inclusion of some studies were resolved by discussion including all authors. An informed consent was obtained from the patient for the publication of the case report.

## 3. Results

### 3.1. Search Results

The search strategy generated 160 papers. Screening of study titles and abstracts revealed 47 articles potentially eligible for inclusion; a further assessment based on the full-text led to the exclusion of 10 papers (Figure 1). The 37 selected studies described 64 adult patients with XX male syndrome [5,6,7,8,9,10,11,12,13,14,15,16,17,18,19,20,21,22,23,24,25,26,27,28,29,30,31,32,33,34,35,36,37,38,39,40,41].

### 3.2. Synthesis of Results 

The clinical data of the patients are summarized in Table 1. Mean (SD) age was 33.14 (11.4) years, mean (SD) weight was 70.3 (10.2) kg and mean (SD) height was 165.3 (7.2) cm. Hair distribution was normal in 29/39 patients (74.3%), gynecomastia was absent in 22/39 cases (56.4%), normal testes volume was reported in 0/14 patients (in 50 patients this data was not available), penis size was normal in 26/32 cases (81.2%), pubic hair had a normal development in 6/7 patients (85.7%), normal erectile function was present in 27/30 cases (90%) and libido was preserved in 20/20 patients (100%).

The hormone profile of the patients is reported in Table 2. The data revealed the common presence of hypergonadotropic hypogonadism. Mean (SD) FSH was 40.00 (19.7) mIU/mL, mean (SD) LH was 26.5 (16.7) mIU/mL and mean (SD) total testosterone was 2.6 (1.4) ng/mL.

The genetic features of the patients are listed in Table 3. All patients had a 46,XX karyotype. The *SRY* gene was detected in 51/57 cases (89.5%) and was absent in 6/57 (10.5%) cases. It was not described in the case reports of four patients. The position of the *SRY* was on the Xp in 97% of the cases.

The patients were stratified by *SRY* into two groups: *SRY*-positive (*SRY*+) and *SRY*-negative (*SRY*–). A statistical analysis was performed in order to find significant clinical or hormonal differences between the two groups (Table 4).

Hair distribution, gynecomastia, testes volume, penis size, pubic hair, erectile dysfunction, libido, FSH, LH, PRL, and E2 were compared: no statistically significant differences (*p* > 0.05) were found between the *SRY*-positive and *SRY*-negative patients for comparable parameters. However, it was not possible to compare all parameters between the two groups because there were few *SRY-negative* cases and limited available data.

## 4. Discussion

The term DSD, introduced by the Chicago Consensus Group in 2005, distinguishes three major groups:DSD with atypical sex chromosome configurations, including Turner syndrome, Klinefelter syndrome, and conditions with 46,XX or 46,XY karyotypes;XY DSD characterized by 46,XY karyotype;XX DSD comprised of conditions characterized by 46,XX karyotype and androgen excess, such as congenital adrenal hyperplasia, P450 oxidoreductase deficiency, or exogenous causes [42].

Male adults with 46,XX and DSD are part of the first of these three groups. Many patients with 46,XX karyotype have external male genitalia [11], but they generally have small testes and may also have abnormalities such as cryptorchidism or hypospadias, azoospermia, hypergonadotropic hypogonadism, varying degrees of gynecomastia, poor facial hair growth, diminished libido, and normal cognitive development [43]. No extra-genital abnormalities were found in all reported cases. However, phenotypes in these patients were different, ranging from severe impairment of the external genitalia to hypospadias and/or cryptorchidism to normal male phenotype. It depends mainly, but not only, on the presence of the sex-determining region Y (*SRY*). In recent years a number of other genes involved in disorders of sex development have also been identified. For instance, *SOX9*, *SOX3*, *DAX1*, *WT1*, *FGF9*, and *SF1* are also involved in the sex determination cascade [44,45]. Therefore, taking the *SRY* gene as the only target gene [3] may not provide enough information, and the clinical manifestations of some of the 46,XX male patients were not consistent with the expression of the *SRY* gene. Regardless, male sex differentiation is mostly dependent on the presence of the *SRY* gene, which drives the primitive gonads into testes formation during early human embryonic development [46]. This belief is held because XX *SRY*+ subjects are generally men with male genitalia, whereas XX *SRY*− subjects have ambiguous genitalia. Nevertheless, there are a few XX *SRY*+ subjects with ambiguous sexual characteristics and, therefore, another Y chromosome gene contributing to complete male sex differentiation has been postulated. In a small number of *SRY*+ cases with ambiguous genitalia, there is a small Y fragment located on inactive X in most metaphases [47,48]. Kolon [44] reports that, in general, the greater the amount of Y chromosome DNA present, the more masculinized the phenotype will be. Male adults with 46,XX and normal external genitalia generally discover their pathology in adulthood because of infertility. Infertility affects approximately 10–15% of all couples worldwide [8]. Approximately 30–40% of infertility cases can be attributed to male factors [49]. In about 15% of male infertility cases, no organic cause can be identified [8]. Idiopathic infertility found in most cases of non-obstructive azoospermia (NOA) or severe oligozoospermia is due to chromosomal abnormalities or mutations of genes involved in sex determination and spermatogenesis [50]. The incidence of cytogenetic abnormality is estimated at 5.8% in infertile men and only 0.5% in the normal population [51].

Diagnosis is based on clinical findings, endocrine testing, and cytogenetic testing [2]. Cryptorchidism is present in 15% and anterior hypospadias in around 10%. Endocrine testing normally reveals hypergonadotropic hypogonadism secondary to testicular failure [2,47]. Cytogenetic testing reveals 46,XX. Approximately 80% of 46,XX DSD individuals are *SRY*+, and around 20% are *SRY*− [2,47]. The investigation is usually based on fluorescence in situ hybridization (FISH) or polymerase chain reaction (PCR) amplification of the *SRY* gene.

There is no literature regarding the comparison between FISH and PCR for *SRY* detection and location. Considering similar studies were performed in different contexts (e.g., detection of *BCR-ABL* fusion gene, detection of translocation *RCC*), we think that FISH and RT-PCR should be used together in order to improve the sensitivity of *SRY* detection and location [42,52].

Various studies indicated that 80–90% of 46,XX males result from a Y to X translocation during meiosis [49,53]. 46,XX males who showed no evidence of Y specific DNA, including *SRY*, were reported [50]. This suggests that testicular development in these males occurred in the absence of the *SRY* gene. On the basis of karyotype analysis and detection of the *SRY* gene, 46,XX male patients can be clinically divided [3] into the *SRY*-positive and the *SRY*-negative groups.

The *SRY* gene is identified as the main gene regulating the testes determination cascade. The most important role of *SRY* is to regulate the *SOX9* expression in Sertoli cell precursors. This pathway, in turn, activates testis-specific genes leading to testis determination [51].

In the absence of *SRY* (*SRY*-negative patients), the male phenotype develops probably from the gain of function in a gene downstream to the *SRY* pathway [29]. *SOX9*, *SOX3*, *DAX1*, *WT1*, *FGF9*, and *SF1* are also involved in the sex determination cascade [54]. While the clinical symptoms of patients often show some degree of heterogeneity [55], usually, the development of genitalia is normal and masculinity signs are obvious in *SRY*+ patients. There is no abnormality in the development of penis and sex psychology as well as erection and ejaculation, and there are almost no significant positive signs except cryptorchidism before puberty. So, it is difficult to find SRY+ male DSD patients before puberty, who are often incidentally found by chromosome check for infertility or poor testicle development. On the contrary, *SRY*− patients could be easily discriminated due to abnormality of genitalia shortly after birth; some patients even show genital ambiguity [55]. Masculinity signs are not clear in *SRY*− patients; especially in adult patients. Breast development and female secondary sex characteristics can be found. For patients with the *SRY*− gene, due to genital ambiguity, patients’ sex psychology and physiological development should be carefully taken into consideration for treatment [3] and the patient may not be able to avoid drug-dependence for maintaining the secondary sex characteristics.

On the basis of our statistical analysis the presence/absence of *SRY* does not seem to affect the clinical and hormonal characteristics of the patients, however the *SRY*− cases are fewer than the *SRY*+ cases. This may have erroneously led to a no statistically significant difference between the two groups.

Individuals with the 46,XX male syndrome may have phenotypic and endocrinologic status that would be expected to be the same as in Klinefelter’s syndrome characterized by 47,XXY. Thus, this syndrome is considered as a variant form of Klinefelter’s syndrome. However, one does not always assume that a 46,XX male will be similar to individuals with Klinefelter’s syndrome [13], which has the characteristics of tall stature, gynecomastia, small testicular volume, a borderline-low intelligence quotient (IQ), and hypergonadotropic hypogonadism. Some 46,XX males do not have gynecomastia, hypogonadism, or even exaggerated LH-releasing hormone (LHRH)-stimulated gonadotropin response [56]. In 46,XX male syndrome, different from Klinefelter’s syndrome, there is short stature which is probably due to translocations of sex chromosomes or to genetic defects that affect growth hormone (GH) activity. Chung-Jung et al. [13] found exaggerated GH response after an insulin-induced hypoglycemic test. They suggested that the short stature in patients with 46,XX male syndrome should not be attributed to the syndrome itself. They speculate possible defects of GH activity as in partial GH insensitivity syndrome that involves genes controlling expression of GH receptor. Anyway, the pathogenesis of 46,XX male DSD is not clear. According to Wang et al. [3], the hypotheses are as follows:The hypothesis of target gene mutation [57]. It supposes the structural gene that determines human gender may be located in autosome, which is regulated by the inhibition of the X chromosome and the activation of the Y chromosome. 46,XX individuals, due to defects in the inhibition of the X chromosome, which results in spontaneous activation of the downstream gene in the absence of the *SRY* gene, transform into 46,XX males.The hypothesis of *SOX9* gene (*SRY* box-related gene 9) overexpression. *SOX* are a large gene family, in which *SOX9* is located in 17q24.3–q25.1 and homologous with *SRY* High-Mobility Group box (HMG)-box as high as 60%. Early studies confirmed that *SOX9* was mainly involved in bone formation and the regulation of Sertoli cell differentiation [58], which was also expressed in the precursor cell of supporting cells with the *SRY* gene expression. In addition, the fact that *SOX9* can be found in spinal animals and mammals, compared with the *SRY* gene which can be found only in mammals, might indicate that *SOX9* was a more ancient gene involved in sex differentiation than the *SRY* gene. Huang et al. [59] reported the case of a *SRY*-negative 46,XX male patient, and chromosome analysis showed the existence of overlapping of the *SOX9* gene. Malki et al. [60] found that prostaglandin D2 (PGD2) could induce the expression of *SOX9* in a normal female rat gonad in vitro. Thus, up-regulation of *SOX9* expression caused by chromosomal abnormalities or mediated by other bypass activation (e.g., PGD2) may result in the occurrence of *SRY*-negative 46,XX male patients.The hypothesis of Xp-Yp translocation. It supposes that the ends of the XY chromosomes’ abnormal exchange (Xp-Yp translocation) occurs during paternal sperm meiosis and results in X-type sperm containing the *SRY* gene, which could lead to 46,XX offspring when combined with eggs. PCR can detect the *SRY* gene, however, only to find it translocated in the X chromosome by FISH. In Brazil, Domenice et al. [61] found that 90% of 46,XX males carried Y chromosome material, including the *SRY* gene, in most cases. It indicated that the Y chromosome fragment containing the *SRY* gene translocation may be an important factor for the occurrence of *SRY*+ 46,XX male patients.

Facing a patient with 46,XX male syndrome, the clinician should keep in mind the possible abnormalities linked to the syndrome and thus perform an adequate screening: an appropriate physical examination should include evaluation and palpation of the genitalia to detect cryptorchidism, hypospadias, penis size, and gynecomastia. It is important to use a validated questionnaire to evaluate erectile dysfunction and diminished libido, such as the International Index of Erectile Function (IIEF); semen analysis is mandatory (even though this is often the reason why they come to the urologist) and so is the karyotype test. An abdominal ultrasound is useful in order to exclude residual Müllerian structures. Genetic and endocrine consultations are necessary to assess the status of the patient who often has hypergonadotropic hypogonadism (Figure 2).

In *SRY*-negative patients, no further instrumental or blood tests are necessary, however, we suggest searching for mutations of other genes involved in the sex determination cascade such as *SOX9*, *SOX3*, *DAX1*, *WT1*, *FGF9*, and *SF1*.

## 5. Conclusions

In conclusion, 46,XX male DSD, characterized by mismatch of genetic, gonadal, and phenotypic sex is quite rare, and due to genetic or chromosomal abnormalities. FISH and PCR technology can quickly and accurately detect information about the *SRY* gene in patients, so as to provide more valuable clinical information. Although chromosomal abnormalities are rarely present in patients with apparently normal external genitalia, they should be considered in urology consultations by adolescents and adults, particularly in the investigation of gynecomastia or infertility. Finally, considering genetic abnormalities, testicular sperm extraction (TESE) is not recommended and these patients should consider adoption or in vitro fertilization with a sperm donor as fertility options.

## Figures and Tables

**Figure 1 medicina-55-00371-f001:**
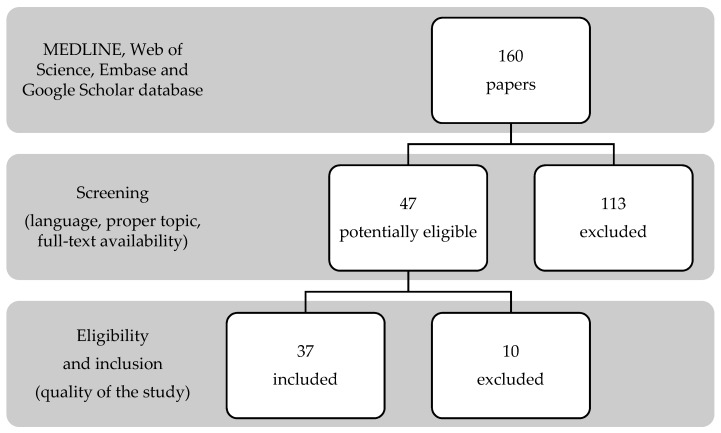
Search strategy according to PRISMA guidelines.

**Figure 2 medicina-55-00371-f002:**
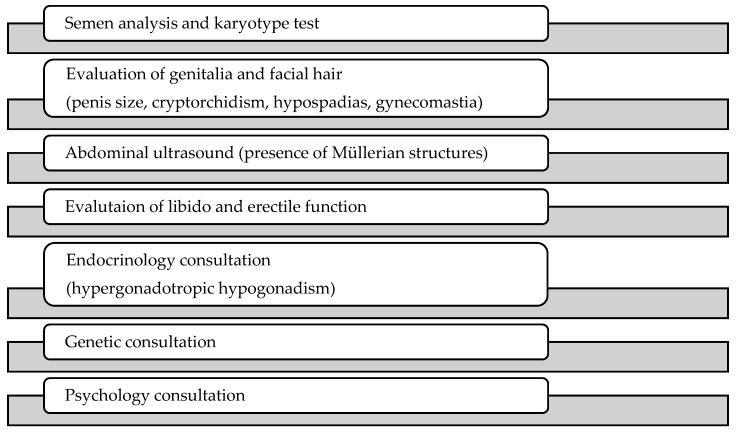
Diagnostic management for 46,XX male syndrome.

**Table 1 medicina-55-00371-t001:** Clinical data of 46, XX male adults.

Authors	Age Years	Weight kg	Height cm	HD	GM	Testes Volume	Penis Size	Pubic Hair	ED	Libido
Guzman et al. [5]	20	76	169	Normal	No	Small	Normal	Normal	No	Normal
Gunes et al. [6]	30	70	155	Poor	Yes	Small	Normal	Normal	No	Normal
Gunes et al. [6]	16	65	152	Normal	No	Small	Normal	Normal	No	Normal
Valetto et al. [7]	35	48	152	Normal	No	NA	Normal	NA	No	Normal
Kim et al. [8]	29	62	165	Normal	No	NA	NA	NA	Yes	NA
Xiao et al. [9]	27	NA	170	NA	NA	NA	Small	NA	No	NA
Queralt et al. [10]	31	58	170	Normal	No	NA	NA	NA	NA	NA
Baziz et al. [11]	44	NA	NA	Normal	Yes	Small	Normal	NA	No	Normal
Tomomasa et al. [12]	25	55	177	Normal	No	Small	NA	Normal	NA	Normal
Chung Jung et al. [13]	17	NA	154	Normal	No	Small	Normal	Normal	No	Normal
Wang et al. [14]	20	NA	NA	Poor	No	Small	Normal	Inverted	No	Normal
Ahsan T et al. [15]	24	NA	NA	Poor	Yes	Small	Normal	Normal	No	Normal
Jain et al. [16]	38	63	162	Normal	Yes	NA	Normal	NA	Yes	NA
Yencilek et al. [17]	26	72	165	Normal	No	NA	Small	NA	No	NA
Pepene et al. [18]	28	65	167	Normal	Yes	Small	Normal	NA	No	Normal
Mustafa et al. [19]	30	75	170	Normal	Yes	NA	Normal	NA	NA	Normal
Majzoub et al. [20]	40	84	175	Normal	No	NA	Normal	NA	No	Normal
Majzoub et al. [20]	31	NA	NA	Normal	No	NA	Normal	NA	No	Normal
Majzoub et al. [20]	35	NA	NA	Poor	Yes	NA	Normal	NA	Yes	Normal
Majzoub et al. [20]	39	74	160	Normal	No	Small	Normal	NA	No	Normal
Majzoub et al. [20]	29	77	181	Normal	No	NA	Normal	NA	No	Normal
Majzoub et al. [20]	32	86	170	Normal	Yes	NA	Normal	NA	No	Normal
Onrat et al. [21]	23	NA	NA	Normal	No	Small	Normal	NA	No	Normal
Hado et al. [22]	76	NA	157	Normal	Yes	NA	NA	NA	No	Normal
Rigola et al. [23]	33	NA	NA	Normal	No	NA	Normal	NA	NA	NA
Dauwerse et al. [24]	61	NA	171	NA	No	Small	Normal	NA	NA	NA
Ryan et al. [25]	40	NA	NA	Poor	No	NA	3.6	NA	No	NA
Gao et al. [26]	NA	NA	163	NA	NA	NA	NA	NA	NA	NA
Gao et al. [26]	NA	NA	163	NA	NA	NA	NA	NA	NA	NA
Gao et al. [26]	NA	NA	162	NA	NA	NA	NA	NA	NA	NA
Gao et al. [26]	NA	NA	161	NA	NA	NA	NA	NA	NA	NA
Gao et al. [26]	NA	NA	158	NA	NA	NA	NA	NA	NA	NA
Gao et al. [26]	NA	NA	162	NA	NA	NA	NA	NA	NA	NA
Gao et al. [26]	NA	NA	162	NA	NA	NA	NA	NA	NA	NA
Gao et al. [26]	NA	NA	161	NA	NA	NA	NA	NA	NA	NA
Gao et al. [26]	NA	NA	160	NA	NA	NA	NA	NA	NA	NA
Gao et al. [26]	NA	NA	160	NA	NA	NA	NA	NA	NA	NA
Gao et al. [26]	NA	NA	161	NA	NA	NA	NA	NA	NA	NA
Rizvi et al. [27]	33	85.8	177	NA	NA	NA	Normal	NA	No	NA
Minor et al. [28]	24	NA	NA	NA	Yes	NA	NA	NA	NA	NA
Rajender et al. [29]	34	64	156	Normal	No	NA	Normal	NA	NA	NA
Tan et al. [30]	NA	NA	176	Normal	Yes	NA	Small	NA	No	NA
Zakharia et al. [31]	65	65	165	NA	Yes	NA	Normal	NA	No	NA
Chiang et al. [32]	33	NA	NA	NA	NA	NA	NA	NA	No	NA
Chiang et al. [32]	34	NA	NA	NA	NA	NA	NA	NA	NA	NA
Chiang et al. [32]	52	NA	NA	NA	NA	NA	Small	NA	NA	NA
Wu et al. [33]	NA	NA	NA	NA	NA	NA	NA	NA	NA	NA
Wu et al. [33]	NA	NA	NA	NA	NA	NA	NA	NA	NA	NA
Wu et al. [33]	NA	NA	NA	NA	NA	NA	NA	NA	NA	NA
Wu et al. [33]	NA	NA	NA	NA	NA	NA	NA	NA	NA	NA
Wu et al. [33]	NA	NA	NA	NA	NA	NA	NA	NA	NA	NA
Chernykh et al. [34]	37	74	160	Poor	Yes	NA	NA	NA	NA	NA
Butler et al. [35]	31	72	169	Normal	Yes	NA	Normal	NA	No	NA
Castineyra et al. [36]	28	NA	180	Normal	Yes	NA	NA	NA	NA	NA
Castineyra et al. [36]	35	NA	170	Normal	NA	Small	NA	NA	NA	NA
Castineyra et al. [36]	28	NA	160	Poor	NA	NA	NA	NA	NA	NA
Castineyra et al. [36]	39	NA	174	Poor	NA	NA	NA	NA	NA	NA
Castineyra et al. [36]	24	NA	172	Normal	NA	NA	NA	NA	NA	NA
Fuse et al. [37]	30	90	172	Normal	No	NA	NA	NA	NA	NA
Pais et al. [38]	29	82	170	Normal	Yes	NA	Small	NA	No	NA
Wegner et al. [39]	35	81	167	Normal	No	NA	Normal	NA	NA	NA
Micic et al. [40]	25	63	171	Poor	No	NA	NA	NA	No	NA
Matthews et al. [41]	27	68	166	Poor	No	NA	NA	NA		NA
Our case	36	74	165	NA	Yes	Small	Normal	NA	No	Normal

NA: not available; HD: hair distribution; GM: gynecomastia; ED: erectile disfunction.

**Table 2 medicina-55-00371-t002:** Hormone profile of 46,XX male adults.

Authors	FSH mIU/mL	LH mIU/mL	TT ng/mL	E2 pg/mL	PRL ng/mL
Guzman et al. [5]	27.9	16.5	2.3	24.8	24.5
Gunes et al. [6]	37.88	18.96	0.51	17.57	17.54
Gunes et al. [6]	41.05	14.55	2.16	32	24.11
Valetto et al. [7]	23.9	17.7	3.06	NA	7.13
Kim et al. [8]	76	41	1.79	NA	NA
Xiao et al. [9]	47	18.7	1.80	NA	14.6
Queralt et al. [10]	62.2	25.8	3.23	17	NA
Baziz et al. [11]	51	11.71	NA	NA	NA
Tomomasa et al. [12]	19.7	10.3	4.28	NA	NA
Chung Jung et al. [13]	25.1	11.4	4.3	30.1	16.8
Wang et al. [14]	77.5	40.75	4.64	NA	15.59
Ahsan T et al. [15]	35	21	1.8	NA	NA
Jain et al. [16]	76.6	36.3	1.20	NA	NA
Yencilek et al. [17]	45.6	48.9	2.70	NA	9.4
Pepene et al. [18]	43.9	25.3	3.33	NA	NA
Mustafa et al. [19]	NA	40.7	2.11	16.6	8.5
Majzoub et al. [20]	38	12	3.35	29	13.6
Majzoub et al. [20]	14	6	1.29	25	3.2
Majzoub et al. [20]	10	23	0.74	5.7	NA
Majzoub et al. [20]	28	15	0.74	NA	NA
Majzoub et al. [20]	13.4	12	2.46	19	NA
Majzoub et al. [20]	29.7	16.9	0.95	NA	12
Onrat et al. [21]	9.95	17.3	0.20	NA	NA
Hado et al. [22]	27.8	21	2.59	NA	NA
Rigola et al. [23]	NA	NA	NA	NA	NA
Dauwerse et al. [24]	13	10	3.25	31.3	NA
Ryan et al. [25]	1	NA	NA	10	NA
Gao et al. [26]	93.6	19.4	3.08	33	17.9
Gao et al. [26]	24.7	14.4	2.77	43	18.5
Gao et al. [26]	NA	NA	1.29	NA	NA
Gao et al. [26]	81.6	27.7	1.37	19.8	22.9
Gao et al. [26]	13.1	3.61	2.44	34	9.67
Gao et al. [26]	54.7	19.4	1.72	27	10.08
Gao et al. [26]	37.1	16.5	3.19	28	9.88
Gao et al. [26]	43	33.9	2.16	22	7.28
Gao et al. [26]	72	34.6	3.36	19.8	10
Gao et al. [26]	49	26.8	1.80	19.8	15.8
Gao et al. [26]	87.7	31.4	5.21	30.5	49.6
Rizvi et al. [27]	46	23	2.07	NA	NA
Minor et al. [28]	55.4	28.4	0.119	NA	NA
Rajender et al. [29]	25.8	15.8	5.8	NA	NA
Tan et al. [30]	21	34	2.63	25	NA
Zakharia et al. [31]	72	61	2.40	NA	16.3
Chiang et al. [32]	46.5	17.6	2.03	NA	27.05
Chiang et al. [32]	54.3	19.6	2.17	NA	8.15
Chiang et al. [32]	64.3	20.2	1.44	NA	16.08
Wu et al. [33]	35.5	13.8	1.95	30.5	4.6
Wu et al. [33]	29.2	12.9	1.55	19.1	3.6
Wu et al. [33]	45.9	25.1	2.56	26.7	7.8
Wu et al. [33]	33.7	22.3	2.41	29.1	10.9
Wu et al. [33]	31.4	19.6	2.01	22.1	7.8
Chernykh et al. [34]	26.9	13.5	2.90	NA	NA
Butler et al. [35]	51	NA	4.77	NA	NA
Castineyra et al. [36]	50	16	3.00	28	14
Castineyra et al. [36]	3.5	6.2	7.00	38	3.4
Castineyra et al. [36]	21	5.2	1.40	19	8.1
Castineyra et al. [36]	6.7	4.2	5.60	30	6.2
Castineyra et al. [36]	45	40	3.00	20	5.4
Fuse et al. [37]	47	60	1.60	NA	NA
Pais et al. [38]	53	45	2.67	NA	NA
Wegner et al. [39]	23.7	37.1	6.30	NA	3.8
Micic et al. [40]	31	18	3.19	47	6.8
Matthews et al. [41]	46	19	2.82	33	9.87
Our case	24.7	9.4	2.7	14	12.2

NA: not available; FSH: follicle-stimulating hormone; LH: luteinizing hormone; TT: total testosterone; E2: estradiol; PRL: prolactin.

**Table 3 medicina-55-00371-t003:** Genetic features of 46,XX male adults.

Authors	Presence of *SRY*	Location of *SRY*
Guzman et al. [5]	+	NA
Gunes et al. [6]	+	NA
Gunes et al. [6]	+	NA
Valetto et al. [7]	NA	NA
Kim et al. [8]	NA	NA
Xiao et al. [9]	−	NA
Queralt et al. [10]	+	NA
Baziz et al. [11]	+	NA
Tomomasa et al. [12]	+	NA
Chung Jung et al. [13]	+	NA
Wang et al. [14]	+	NA
Ahsan T et al. [15]	NA	NA
Jain et al. [16]	+	NA
Yencilek et al. [17]	NA	NA
Pepene et al. [18]	+	NA
Mustafa et al. [19]	−	NA
Majzoub et al. [20]	+	NA
Majzoub et al. [20]	+	NA
Majzoub et al. [20]	+	NA
Majzoub et al. [20]	−	NA
Majzoub et al. [20]	+	NA
Majzoub et al. [20]	+	NA
Onrat et al. [21]	+	NA
Hado et al. [22]	+	NA
Rigola et al. [23]	+	Xp
Dauwerse et al. [24]	+	16q
Ryan et al. [25]	−	NA
Gao et al. [26]	+	Xp
Gao et al. [26]	+	Xp
Gao et al. [26]	+	Xp
Gao et al. [26]	+	Xp
Gao et al. [26]	+	Xp
Gao et al. [26]	+	Xp
Gao et al. [26]	+	Xp
Gao et al. [26]	+	Xp
Gao et al. [26]	+	Xp
Gao et al. [26]	+	Xp
Gao et al. [26]	+	Xp
Rizvi et al. [27]	+	Xp
Minor et al. [28]	+	Xp
Rajender et al. [29]	−	NA
Tan et al. [30]	NA	NA
Zakharia et al. [31]	NA	NA
Chiang et al. [32]	+	Xp
Chiang et al. [32]	+	Xp
Chiang et al. [32]	−	NA
Wu et al. [33]	+	Xp
Wu et al. [33]	+	Xp
Wu et al. [33]	+	Xp
Wu et al. [33]	+	Xp
Wu et al. [33]	+	Xp
Chernykh et al. [34]	+	Xp
Butler et al. [35]	+	Xp
Castineyra et al. [36]	+	Xp
Castineyra et al. [36]	+	Xp
Castineyra et al. [36]	+	Xp
Castineyra et al. [36]	+	Xp
Castineyra et al. [36]	+	Xp
Fuse et al. [37]	+	Xp
Pais et al. [38]	+	Xp
Wegner et al. [39]	+	Xp
Micic et al. [40]	NA	NA
Matthews et al. [41]	+	Xp
Our case	+	Xp

NA: not available; *SRY*: sex-determining region Y; Xp: short arm of X chromosome.

**Table 4 medicina-55-00371-t004:** Comparison of *SRY*-positive and *SRY*-negative patients.

Patients	HD% Normal	GM% No	Penis size% Normal	ED% No	FSH mIU/mL Mean (SD)	LH mIU/mL Mean (SD)	TT ng/mL Mean (SD)	PRL ng/mL Mean (SD)
*SRY*+	73.3	55.2	90.9	90.0	38.63 (20.85)	29.40 (19.09)	2.64 (1.49)	12.93 (9.32)
*SRY*−	66.7	57.1	60.0	85.7	51.74 (28.14)	21.63 (9.56)	2.49 (1.79)	14.27 (4.07)
	*p =* 0.422	*p =* 0.596	*p =* 0.144	*p =* 0.823	*p =* 0.195	*p =* 0.326	*p =* 0.819	*p =* 0.781

HD: hair distribution; GM: gynecomastia; ED: erectile disfunction; FSH: follicle-stimulating hormone; LH: luteinizing hormone; TT: total testosterone; PRL: prolactin.

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
