# Peer review of "46,XX Testicular Disorder of Sex Development (DSD): A Case Report and Systematic Review"

_1010-660X, 2019, doi:10.3390/medicina55070371_

Round 1

Reviewer 1 Report

1. This review is valuable to the guidance of 46,XX, male patients. However, the difference between the 46, XX, SRY positive and SRY negative part in hormone profile and related clinical data should be amplified by to further expound the difference of consultation strategy.

2. What is the role of SRY present in typical 46,XX, male patients patients?

3. The presence of SRY was detected by both PCR and FISH, however many cases did’t show the location of SRY translacation. The location of SRY if will affect the clinical presentation. Which experiment is considered gold standard for 46, XX, SRY positive patient? Please give some discussion on this issue.

4. Several errors presented in the reference section, please correct.

Author Response

RESPONSE TO REVIEWER 1 COMMENTS

Thank you for your comments and suggestions.

POINT 1: This review is valuable to the guidance of 46,XX, male patients. However, the difference between the 46, XX, SRY positive and SRY negative part in hormone profile and related clinical data should be amplified by to further expound the difference of consultation strategy.

RESPONSE 1: The patients were stratified by SRY into two groups: SRY-positive and SRY-negative. A statistical analysis were performed in order to find clinical or hormonal significant differences between the two groups.

Hair distribution, gynecomastia, testes volume, penis size, pubic hair, erectile dysfunction, libido, FSH, LH, PRL and E2 were compared: no statistically significant differences (p > 0.05) were found between the SRY-positive and SRY-negative patients for comparable parameters. In other words, the presence / absence of SRY does not seem to affect the characteristics of the patients.

In SRY-negative patients, no further instrumental or blood tests are necessary, however, we suggest to search for mutations of other genes involved in the sex determination cascade such as SOX9, SOX3, DAX1, WT1, FGF9 and SF1.

POINT 2: What is the role of SRY present in typical 46,XX, male patients patients?

RESPONSE 2: The SRY gene has been identified as the main gene regulating the testes determination cascade. The most important role of SRY is to regulate the SOX9 expression in Sertoli cell precursors. This pathway, in turn, activates testis-specific genes leading to testis determination.

(She, Z. Y., Yang, W. X. Sry and SoxE genes: How they participate in mammalian sex determination and gonadal development? Seminars in Cell & Developmental Biology. 2017 Mar, 63, 13–22.)

In the absence of SRY (SRY-negative patients), the male phenotype develops probably from the gain of function in a gene downstream to SRY pathway.

(Singh Rajender, Vutukuri Rajani, Nalini J.Gupta, Baidyanath Chakravarty, Lalji Singh and Kumarasamy Thangaraj. SRY-negative 46,XX male with normal genitals, complete masculinization and infertility. MHR: Basic science of reproductive medicine, Volume 12, Issue 5, May 2006, Pages 341–346)

SOX9, SOX3, DAX1, WT1, FGF9 and SF1 are also involved in the sex determination cascade

(Mizuno K, Kojima Y, Kamisawa H, Moritoki Y, Nishio H, Kohri K, Hayashi Y: Gene expression profile during testicular development in patients with SRY-negative 46,XX testicular disorder of sex development. Urology 2013, 82(6):1453. e1-e7.).

POINT 3: The presence of SRY was detected by both PCR and FISH, however many cases did’t show the location of SRY translacation. The location of SRY if will affect the clinical presentation. Which experiment is considered gold standard for 46, XX, SRY positive patient? Please give some discussion on this issue.

RESPONSE 3: There is not literature regarding the comparison between FISH and PCR for the SRY detection and location. Taking into account of similar studies performed in different contexts (e.g. detection of BCR-ABL fusion gene, detection of translocation RCC), we think that FISH and RT-PCR should be used togheter in order to improve the sensitivity of SRY detection and location.

(Cox MC1, Maffei L, Buffolino S, Del Poeta G, Venditti A, Cantonetti M, Aronica G, Aquilina P, Masi M, Amadori S.  A Comparative Analysis of FISH, RT-PCR, and Cytogenetics for the Diagnosis of bcr-abl Positive Leukemias. Am J Clin Pathol. 1998 Jan;109(1):24-31.

Lee HJ1,2, Shin DH1,2, Noh GY1, Kim YK1, Kim A1, Shin N1, Lee JH1, Choi KU1, Kim JY1, Lee CH1, Sol MY1, Rha SH3, Park SW4. Combination of immunohistochemistry, FISH and RT-PCR shows high incidence of Xp11 translocation RCC: comparison of three different diagnostic methods. Oncotarget. 2017 May 9;8(19):30756-30765.)

POINT 4: Several errors presented in the reference section, please correct.

RESPONSE 4: The reference section has been revised

Reviewer 2 Report

The authors describe a case and a literature review of 46,XX males. This is an interesting and rare disorder.  The literature review gives interesting information. However, the authors can present the data, and analyze the data, in a more informative way.

It would be very interesting to know if there are group differences based on the genetic findings – i.e. comparing those positive for SRY with those that do not have an SRY

The karyotype does not have to be listed in the table since that is the basis for inclusion – more interesting to have SRY yes no

The manuscript would benefit from describing the patients as two groups – with SRY or not, both in the tables and the statistics.

Please present the group data for the whole group and the two groups separately.

This can possibly also be presented in a table with means and SDS. If there are significant group differences this should be presented as well, or mentioned that there are not.

I suggest that the column for free testosterone Is omitted in the table since there is only one patient with this information – the case the authors are presenting in the text

Erectile dysfunction – should be yes, no, or NA for all patients (see last one)

Please define what you mean with poor hair distribution.

Author Response

RESPONSE TO REVIEWER 2 COMMENTS

Thank you for your comments and suggestions.

POINT 1: It would be very interesting to know if there are group differences based on the genetic findings – i.e. comparing those positive for SRY with those that do not have an SRY.

POINT 3: The manuscript would benefit from describing the patients as two groups – with SRY or not, both in the tables and the statistics.

Please present the group data for the whole group and the two groups

separately.

This can possibly also be presented in a table with means and SDS. If there

are significant group differences this should be presented as well, or

mentioned that there are not.

RESPONSE 1 & 3: The patients were stratified by SRY into two groups: SRY-positive and SRY-negative. A statistical analysis were performed in order to find clinical or hormonal significant differences between the two groups.

Hair distribution, gynecomastia, testes volume, penis size, pubic hair, erectile dysfunction, libido, FSH, LH, PRL and E2 were compared: no statistically significant differences (p > 0.05) were found between the SRY-positive and SRY-negative patients for comparable parameters.  In other words, the presence / absence of SRY does not seem to affect the characteristics of the patients.

POINT 2: The karyotype does not have to be listed in the table since that is the basis for inclusion – more interesting to have SRY yes no.

RESPONSE 2: That column of karyotype has been deleted, the column of SRY is present.

POINT 4: I suggest that the column for free testosterone Is omitted in the table since there is only one patient with this information – the case the authors are presenting in the text.

RESPONSE 4: The column for free testosterone has been deleted, this data has been presented in the text

POINT 5: Erectile dysfunction – should be yes, no, or NA for all patients (see last one)

RESPONSE 5: The presence of erectile dysfunction in the table has been modified in yes, no or NA for the all cases.

POINT 6: Please define what you mean with poor hair distribution.

RESPONSE 6: The definition has been added in the text

Round 2

Reviewer 2 Report

The authors have made some changes according to my previous comments.

However, some issues still remain.

The English language still needs to be corrected. I don’t think I am the right person for this since English is not my native language.

Line 20- man should be male – is this how the search was performed?

Line29. Validated

Line 30. Should be mandatory

Line 31 Genetic and endocrine consultations… (no A)

Line 50 referred should be substituted for another work – unclear to me what the authors mean

And similarly in the following text, will not continue through correcting through the text

The way the information is presented is the same: In Table X ….

The information would be more appealing and interesting if the findings are presented and then referred to in the following way - (Table X)

Lines 121 -124, when data is presented as mean and (SD) it is sufficient to state this for the first results reported.

The number of individuals with SRY present differ between the text (36) and the table (more than 50). This has to be corrected.

Author Response

RESPONSE TO REVIEWER 2 COMMENTS

Thank you for your comments and suggestions.

POINT 1: The English language still needs to be corrected. I don’t think I am the right person for this since English is not my native language.

Line 20- man should be male – is this how the search was performed?

Line29. Validated

Line 30. Should be mandatory

Line 31 Genetic and endocrine consultations… (no A)

Line 50 referred should be substituted for another work – unclear to me what the authors mean

And similarly in the following text, will not continue through correcting through the text.

RESPONSE 1: The grammar of the whole text has been revised and corrected. In line 20 there was a mistake, "male" is the correct word as well as the search term used.

POINT 2: The way the information is presented is the same: In Table X ….

The information would be more appealing and interesting if the findings are presented and then referred to in the following way - (Table X)

Lines 121 -124, when data is presented as mean and (SD) it is sufficient to state this for the first results reported.

RESPONSE 2: All our findings have been included in a new Table

POINT 3: The number of individuals with SRY present differ between the text (36) and the table (more than 50). This has to be corrected.

RESPONSE 3: The data have been corrected according to the table

Medicina EISSN 1010-660X Published by MDPI AG, Basel, Switzerland RSS E-Mail Table of Contents Alert
Back to Top